# Evaluation of the Ability to Form Biofilms in KPC-Producing and ESBL-Producing *Klebsiella pneumoniae* Isolated from Clinical Samples

**DOI:** 10.3390/antibiotics12071143

**Published:** 2023-07-02

**Authors:** Carolina Sabença, Eliana Costa, Sara Sousa, Lillian Barros, Ana Oliveira, Sónia Ramos, Gilberto Igrejas, Carmen Torres, Patrícia Poeta

**Affiliations:** 1MicroART-Antibiotic Resistance Team, Department of Veterinary Sciences, University of Trás-os-Montes and Alto Douro, 5000-801 Vila Real, Portugal; anacarolina@utad.pt; 2Department of Genetics and Biotechnology, University of Trás-os-Montes and Alto Douro, 5000-801 Vila Real, Portugal; gigrejas@utad.pt; 3Functional Genomics and Proteomics Unit, University of Trás-os-Montes and Alto Douro, 5000-801 Vila Real, Portugal; 4Associated Laboratory for Green Chemistry, University NOVA of Lisbon, 1099-085 Caparica, Portugal; 5Hospital Centre of Trás-os-Montes and Alto Douro, Clinical Pathology Department, 5000-508 Vila Real, Portugal; ecsvalente@chtmad.min-saude.pt (E.C.); sisousa@chtmad.min-saude.pt (S.S.); 6Centro de Investigação de Montanha (CIMO), Instituto Politécnico de Bragança, Campus de Santa Apolónia, 5300-253 Bragança, Portugal; lillian@ipb.pt; 7Laboratório Associado para a Sustentabilidade e Tecnologia em Regiões de Montanha (SusTEC), Instituto Politécnico de Bragança, Campus de Santa Apolónia, 5300-253 Bragança, Portugal; 8Egas Moniz Center for Interdisciplinary Research (CiiEM), Egas Moniz School of Health and Science, 2829-511 Caparica, Portugal; ana.dermatology@gmail.com; 9Faculty of Veterinary Medicine, Centro Universitário de Lisboa, Campo Grande, 376, 1749-024 Lisbon, Portugal; 10Area Biochemistry and Molecular Biology, University of La Rioja, 26006 Logroño, Spain; carmen.torres@unirioja.es; 11CECAV—Veterinary and Animal Research Centre, University of Trás-os-Montes and Alto Douro, 5000-801 Vila Real, Portugal; 12Veterinary and Animal Research Centre, Associate Laboratory for Animal and Veterinary Science (AL4AnimalS), 5000-801 Vila Real, Portugal

**Keywords:** *Klebsiella pneumoniae*, extended-spectrum β-lactamase (ESBL), *Klebsiella pneumoniae* carbapenemase (KPC), biofilm, antimicrobial resistance

## Abstract

The appearance of *Klebsiella pneumoniae* strains producing extended-spectrum β-lactamase (ESBL), and carbapenemase (KPC) has turned into a significant public health issue. ESBL- and KPC-producing *K. pneumoniae*’s ability to form biofilms is a significant concern as it can promote the spread of antibiotic resistance and prolong infections in healthcare facilities. A total of 45 *K. pneumoniae* strains were isolated from human infections. Antibiograms were performed for 17 antibiotics, ESBL production was tested by Etest ESBL PM/PML, a rapid test was used to detect KPC carbapenemases, and resistance genes were detected by PCR. Biofilm production was detected by the microtiter plate method. A total of 73% of multidrug resistance was found, with the highest resistance rates to ampicillin, trimethoprim–sulfamethoxazole, cefotaxime, amoxicillin-clavulanic acid, and aztreonam. Simultaneously, the most effective antibiotics were tetracycline and amikacin. *bla_CTX-M_*, *bla_TEM_*, *bla_SHV_*, *aac(3)-II*, *aadA1*, *tetA*, *cmlA*, *catA*, *gyrA*, *gyrB*, *parC*, *sul1*, *sul2*, *sul3*, *bla_KPC_*, *bla_OXA_*, and *bla_PER_* genes were detected. Biofilm production showed that 80% of *K. pneumoniae* strains were biofilm producers. Most ESBL- and KPC-producing isolates were weak biofilm producers (40.0% and 60.0%, respectively). There was no correlation between the ability to form stronger biofilms and the presence of ESBL and KPC enzymes in *K. pneumoniae* isolates.

## 1. Introduction

*Klebsiella pneumoniae* is a Gram-negative bacterium that is commonly found in the human gut and can cause severe infections in different parts of the body, especially in people with debilitated immune systems [1]. The emergence of extended-spectrum β-lactamase (ESBL) and *Klebsiella pneumoniae* carbapenemase (KPC) producing strains of *K. pneumoniae* has become a major public health concern, as these strains are resistant to many antibiotics and pose a significant challenge in the treatment of infections [2].

ESBLs are enzymes produced by bacteria that break down the β-lactam antibiotics, such as penicillins, cephalosporins, and carbapenems. Since these antibiotics are commonly prescribed to treat these bacterial infections, the treatment of ESBL-producing *K. pneumoniae* becomes a challenge [3]. KPCs are a type of carbapenemase enzyme that can break down carbapenem antibiotics, which are often used as a last-resort treatment for antibiotic-resistant infections. This means that KPC-producing *K. pneumoniae* can be resistant to almost all available antibiotics [4].

One of the important virulence factors of *K. pneumoniae* is its capacity to form biofilms [5]. Biofilms are communities of microorganisms that adhere to a surface and produce a matrix of extracellular polymeric substances (EPS). The EPS matrix protects the microorganisms from environmental stress, such as antibiotic treatment, making biofilm-associated infections difficult to treat [6]. *K. pneumoniae* is known to form biofilms on various surfaces in healthcare settings, such as medical devices and surfaces in hospitals [7,8].

Studies have shown that ESBL-producing *K. pneumoniae* strains have a higher ability to form biofilms compared to non-ESBL-producing strains [8,9]. This is because ESBLs provide a survival advantage to the bacteria, allowing them to resist the effects of antibiotics and persist in the host [10]. The EPS matrix also provides an environment for horizontal gene transfer, which can facilitate the spread of antibiotic resistance [11,12].

KPC-producing *K. pneumoniae* strains can produce biofilms [13], and this ability has been associated with high levels of antibiotic resistance. This can result in persistent infections that are difficult to treat and can lead to increased morbidity and mortality [14].

There are several methods for evaluating the ability of bacteria to form biofilms. One commonly used method is the microtiter plate assay, in which bacteria are grown in a microtiter plate with a medium that promotes biofilm formation. After a period of incubation, the wells are stained with crystal violet, which binds to the biofilm matrix. The stained biofilm can then be visualized and quantified using microscopy or spectrophotometry [15].

In addition to evaluating biofilm formation, it is also important to understand the genetic mechanisms underlying biofilm formation in *K. pneumoniae*. Several genes and regulatory pathways have been implicated in biofilm formation in *K. pneumoniae*, including the type 1 fimbriae and curli systems, as well as the quorum sensing and cyclic dimeric guanosine monophosphate signaling pathways. Understanding these mechanisms may provide insights into new targets for the development of therapies to prevent or disrupt biofilm formation in *K. pneumoniae* [16,17].

The formation of biofilms by ESBL- and KPC-producing *K. pneumoniae* is a major concern, as it can increase the spread of antibiotic resistance and lead to the persistence of infections in healthcare settings. It is important to understand the mechanisms of biofilm formation and to develop approaches to control and prevent these infections [13].

In conclusion, ESBL- and KPC-producing *K. pneumoniae* are a main concern in the healthcare setting due to their ability to form biofilms and resist antibiotics. The formation of biofilms by these strains can increase the spread of antibiotic resistance and make it difficult to treat infections. Therefore, understanding the mechanisms of biofilm formation will help to reduce the spread of antibiotic resistance and improve patient outcomes [18].

## 2. Materials and Methods

### 2.1. Bacterial Isolates and Identification

Forty-five *Klebsiella pneumoniae* strains were isolated at the hospital center of Trás-os-Montes and Alto Douro between 7 December 2021, and 11 August 2022. The identification of the isolates was conducted by VITEK^®^ 2 Compact (BioMérieux, Auvergne-Rhône-Alpes, France). The strains used in this study were isolated from urinary infections, bacteremia episodes, pulmonary infections, and wounds.

### 2.2. Antimicrobial Resistance Profile

The phenotypic resistance characterization of the isolates was performed by the Kirby–Bauer disk diffusion method by following EUCAST standards (2022) [19], except for ceftazidime, cefotaxime, tetracycline, and nalidixic acid, which followed CLSI standards (2021) [20]. Extended-spectrum β-lactamases (ESBL) production was tested to the Etest ESBL PM/PML (BioMérieux, Auvergne-Rhône-Alpes, France). A single immunochromatography rapid test was used to detect KPC carbapenemases (RESIST-3 O.K.N. K-SeT, Coris BioConcept, Gembloux, Belgium).

A total of 17 antibiotics were tested: Tetracycline (TE) (30 μg), Ceftazidime (CAZ) (10 μg), Cefepime (FEP) (30 μg), Amikacin (AK) (30 μg), Gentamicin (CN) (10 μg), Ciprofloxacin (CIP) (5 μg), Trimethoprim–Sulfamethoxazole (SXT) (1.25/23.75 μg), Cefoxitin (FOX) (30 μg), Imipenem (IMP) (10 μg), Meropenem (MEM) (10 μg), Aztreonam (ATM) (30 μg), Amoxicillin-clavulanic acid (AUG) (20–10 μg), Chloramphenicol (CHL) (30 μg), Ampicillin (AMP) (10 μg), Cefotaxime (CTX) (5 μg), Ertapenem (ERT) (10 μg), and Nalidixic acid (NA) (30 μg).

### 2.3. Molecular Characterization of K. pneumoniae Isolates

#### 2.3.1. DNA Extraction

*K. pneumoniae* isolates were cultured in Brain Heart Infusion agar for 24 h at 37 °C. A loop full of bacteria was suspended in 500 µL of distilled water, and the DNA was extracted by the boiling method [21].

#### 2.3.2. Detection by PCR of Resistance Genes

*K. pneumoniae* strains were screened for multiple resistance genes according to their resistance profile. The presence of genes encoding TEM, SHV, OXA, and CTX-M β-lactamases was studied by PCR [22]. The *tetA*, *tetB*, *aac(3)-II*, *aac(3)-IV*, *aadA1*, *sul1*, *sul2*, *sul3*, *dfrA*, *cmlA*, *floR*, *catA*, *gyrA*, *gyrB*, and *parC* antimicrobial resistance genes were also studied using PCR [23]. The presence of genes encoding KPC, NDM, OXA-48, IMP, VIM, VIM-2, SPM, and PER carbapenemases was also studied by PCR [24,25,26,27,28]. Positive and negative controls from the University of Trás-os-Montes and Alto Douro (Portugal) strain collection were included in all PCR assays.

### 2.4. Biofilm Formation and Biomass Quantification

The bacterial adhesion of all isolates was assessed using a microtitre plate-based assay as previously described with some modifications [15]. Briefly, a few colonies of each isolate were transferred from fresh cultures to tubes with 3 mL of Tryptic Soy Broth (TSB) and incubated at 37 °C for 24 h. Following incubation, the number of cells in each culture was quantified and adjusted to 0.5 McFarland (1.5 × 10^8^ CFU/mL), and 100 μL of each bacterial suspension was transported to a 96-well microtiter plate. *Pseudomonas aeruginosa* ATCC^®^ 27,853 is a recognized biofilm-forming strain used as a positive control in biofilm assay. Sterile TSB was incorporated as a negative control. The microplates were incubated for 24 h at 37 °C. After incubation, bacterial cells in suspension were removed by turning the microplates over, and they were washed twice with distilled water. This step helps remove stray cells and media components that may be stained in the next step, significantly reducing background staining. The plates were then allowed to dry at room temperature for 15 min. Then, 125 µL of methanol (Scharlau, Barcelona, Spain) was added to each well and incubated for 15 min to fix the biofilm. Methanol was removed, the plates were allowed to dry at room temperature for 10–15 min, and 125 µL of Crystal Violet (CV) at 1% (*v*/*v*) (Liofilchem, Roseto degli Abruzzi, Italy) was added to each well. After incubation, the CV solution was removed, and the microplates were washed 3–4 times with distilled water. Subsequently, the plates were vigorously dried on a stack of paper towels to remove all excess cells and stains and were left to dry overnight. 

To quantify the biofilm biomass, 125 µL of acetic acid 30% (*v*/*v*) was added to each well of the microtiter plate to solubilize the CV. After incubation at room temperature for 10–15 min, optical density was read at 630 nm (OD630 nm) [13] using a microplate reader BioTek ELx808U (BioTek, Winooski, VT, USA). The results were interpreted as weak, moderate, and strong biofilm producers. The optical density cut-off value (ODc) was determined by arithmetically averaging the OD of the negative control wells and adding a standard deviation of +3. Samples with an OD higher than the ODc were considered positive, whereas those with a lower optical density than the cut-off value were considered negative. Strains were classified using the following criteria: OD ≤ ODc non-biofilm producer; ODc < OD ≤ 2 × ODc, weak biofilm producer; 2 × ODc < OD ≤ 4 × ODc, moderate biofilm producer; OD > 4 × ODc, strong biofilm producer [29].

## 3. Results

### 3.1. Bacterial Isolates and Identification

This study was conducted on 45 *Klebsiella pneumoniae* strains, including 15 ESBL producers, 15 KPC producers, and 15 non-β-lactamase-producers. Based on the type of specimen, we have 77.8% isolates from urinary infections, 11.1% from bacteremia episodes, 6.7% from pulmonary infections, and 4.4% from wounds. The distribution of isolates is presented in Figure 1. 

### 3.2. Antimicrobial Resistance Profile

The phenotypic profile of the 45 *Klebsiella pneumoniae* isolates demonstrated that 73% of the strains were multi-resistant, showing resistance to three or more antibiotic classes (Table 1). The majority of *K. pneumoniae* isolates showed resistance to ampicillin (*n* = 45), trimethoprim–sulfamethoxazole (*n* = 31), cefotaxime (*n* = 30), amoxicillin-clavulanic acid (*n* = 30), and aztreonam (*n* = 30). At the same time, the most effective antibiotics were tetracycline (*n* = 8) and amikacin (*n* = 7) (Table 2).

None of the isolates was susceptible to all antibiotics tested, but two of them only showed resistance to ampicillin (HS39 and HS63). The most resistant strains presented resistance to 15 different antibiotics (13.3%; *n* = 6), and the most susceptible ones showed resistance to only one antibiotic, ampicillin (4.4%; *n* = 2). Among the most resistant strains, we verified that all of them were KPC-producing *K. pneumoniae*. This suggests that high rates of resistance to commonly used antimicrobial agents are speculated to be associated with KPC production. Another result that strengthens this suggestion is the absence of either ESBL or KPC enzymes in the strains that showed resistance to only one antibiotic (Table 1).

We also verified that some *K. pneumoniae* isolates had the same phenotype profile. Thus, the most common was AMP—CN with 15.6% (*n* = 7), followed by AMP—AUG—FOX—CAZ—CTX—FEP—ATM—MEM—ERT—IMP—CN—CIP—NA—SXT—CHL with 11.1% (*n* = 5) also being the most resistant strains, AMP—AUG—CAZ—CTX—FEP—ATM—CN—CIP—NA—SXT—CHL with 8.8% (*n* = 4), AMP—AUG—FOX—CAZ—CTX—FEP—ATM—ERT—IMP—CN—CIP—NA—SXT—CHL with (4.4%) (*n* = 2), AMP—AUG—CAZ—CTX—FEP—ATM—CIP—NA—SXT—CHL with 4.4% (*n* = 2), and AMP—AUG—CAZ—CTX—FEP—ATM—CN—TET—CIP—NA—SXT with 4.4% (*n* = 2) (Table 1).

### 3.3. Detection of Resistance Genes

In our study, the *bla_CTX-M_*, *bla_TEM_*, and *bla_SHV_* genes were screened against the isolates resistant to penicillins and cephalosporins (*n* = 45). In all ESBL strains, we detected at least one of the β-lactamase genes. The *bla_CTX-M_* was detected in 30 isolates (66.7%), *bla_TEM_* in 24 isolates (53.3%), and *bla_SHV_* in 43 isolates (95.6%). In the majority of the isolates, these genes were present in combinations between them, with *bla_CTX-M_* + *bla_SHV_* being the most frequently found among all samples (*n* = 17, 37.8%), followed by *bla_TEM_* + *bla_SHV_* (*n* = 12, 26.7%) and *bla_CTX-M_* + *bla_TEM_* + *bla_SHV_* (*n* = 12, 26.7%). The *bla_TEM_* alone was not detected, *bla_CTX-M_* alone was verified in one isolate, and *bla_SHV_* alone was present in two samples. Only in one isolate (HS39) was the amplification of these genes was not verified (Table 1). 

Regarding aminoglycoside resistance genes, we tested three different genes: *aac(3)-II*, *aac(3)-IV*, and *aadA1*. The *aac(3)-IV* was not detected. The *aac(3)-II* gene was detected in 27 strains, and the *aadA1* gene was detected in 32 strains. We verified that in the 27 strains where we detected the *aac(3)-II* gene, the *aadA1* gene was also present; however, we verified in 5 strains the presence of the *aadA1* gene alone. In only one strain (HS125), none of the genes tested was detected (Table 1).

The *tetA* and *tetB* genes were tested in the *K. pneumoniae* samples that were resistant to tetracycline. The *tetA* gene was detected in four isolates, and *tetB* was not detected in the tetracycline-resistant samples. In four tetracycline-resistant samples, neither of the genes was detected (Table 1).

The amplification of the genes conferring resistance to chloramphenicol tested in this study was *cmlA*, *floR*, and *catA*. A total of 20 of the isolates showed resistance to this antibiotic, and we could only detect the presence of the *cmlA* and *floR* in one strain (HS97). In the rest of them, none of the genes were amplified (Table 1). 

The *sul1*, *sul2*, *sul3*, and *dfrA* genes were tested in the trimethoprim–sulfamethoxazole-resistant strains. The sul1 was detected in 5 isolates, the *sul2* gene was detected in 24 isolates, and the *sul3* gene was detected only in 1 isolate. Only two strains showed combinations of these genes: *sul1* and *sul2* were verified in the isolate HS154, and *sul2* and *sul3* were present in the isolate HS97. The *dfrA* gene was not detected in any isolate, and in three of them (HS102, HS72, and HS85), none of these genes were detected (Table 1). 

The *gyrA*, *gyrB*, and *parC* genes, target genes for mutations in the quinolone resistance-determining regions, were detected in some of the isolates. The *gyrA* gene was verified in 9 strains, the *gyrB* gene in 22 strains, and the *parC* in 8 strains. Further investigation is needed to identify the specific mutations that confer resistance to the quinolones antibiotics (Table 1). 

Regarding the genes responsible for conferring resistance to carbapenem antibiotics, we tested the *bla_KPC_*, *bla_NDM_*, *bla_OXA_*, *bla_OXA-48_*, *bla_IMP_*, *bla_VIM_*, *bla_VIM-2_*, *bla_SPM_*, and *bla_PER_* genes only in the KPC-producing strains. So, the presence of the *bla_KPC_* gene in all KPC-producing isolates confirms the phenotypic detection test. Concerning the other genes, we only detected the *bla_OXA_* gene in 11 isolates, and the *bla_PER_* gene was present in one isolate (HS160) (Table 1). 

### 3.4. Detection of Biofilm

Among total (*n* = 45) *K. pneumoniae* strains, 36 (80.0%) were confirmed as biofilm producers, of which 26 isolates (57.8%) were weak producers, 9 (20.0%) were moderate, and 1 (2.2%) was strong biofilm producer (Table 3).

### 3.5. Biofilm Production among Clinical Specimen

Among the different clinical specimen isolates, we could verify that 80% of the isolates from urinary infections were able to produce biofilms, the majority being weak producers (*n* = 21). The same percentage was demonstrated in the bacteremia isolates with the same amount of weak (*n* = 2) as moderate biofilm producers (*n* = 2). All of the pulmonary infection isolates were able to produce biofilms (*n* = 3), being classified as weak biofilm producers. One of the two wound isolates was capable of producing biofilm, this being a moderate biofilm producer (Table 3). 

### 3.6. Biofilm Production among ESBL-, KPC- and Non-β-Lactamase-Producing Klebsiella pneumoniae

Among total (*n* = 15) ESBL-producing strains, 11 (73%) were confirmed as biofilm producers, of which 6 isolates (40.0%) were weak producers, 4 (26.7%) were moderate and 1 (6.7%) was strong biofilm producer. Similarly, regarding KPC-producing strains (*n* = 15), 11 (73%) were confirmed as biofilm producers, of which 9 isolates (60.0%) were weak producers, and the remaining 2 (13.3%) were moderate biofilm producers. Between non-β-lactamase-producing strains (*n* = 15), 14 (93%) were confirmed as biofilm producers, of which 11 isolates (73.3%) were weak producers, and the remaining 3 (20.0%) were moderate biofilm producers (Table 3). 

We verified that the most resistant strains were between the non-producers and the weak biofilm producers. On the contrary, the least resistant strains were both moderate biofilm producers. 

## 4. Discussion

This study was carried out to evaluate the ability of ESBL-producing and KPC-producing *Klebsiella pneumoniae* strains, obtained at the hospital center of Trás-os-Montes and Alto Douro, to form stronger biofilms than *K. pneumoniae* strains without these enzymes. It is important to note that due to privacy and ethical constraints, patient data, including clinical information, were not accessible for this study. As a result, an analysis investigating the relationship between the bacteriological characteristics of the *K. pneumoniae* strains and the patients was not feasible. Therefore, the comprehensive understanding of the findings may be restricted by the absence of patient data. Due to their capacity to generate β-lactamase enzymes and biofilms, *Klebsiella pneumoniae* shows resistance to numerous antibiotics [30,31]. Since many β-lactamase genes are located on mobile genetic elements controlled by plasmids, the resistant strains are spreading quickly, causing elevated death rates, illness, and healthcare expenses [32]. The simultaneous expression of multiple β-lactamase genes in an organism can worsen the drug-resistance problem, reducing available treatment options [33]. Hence, identifying these factors and their connection to drug resistance is crucial in diagnostic labs.

*Klebsiella* spp. are microorganisms that can cause multiple infections, such as urinary tract infections, pneumonia, and blood and wound infections. In this work, we isolated 45 *K. pneumoniae*, the majority from urinary infections (77.8%), followed by bacteremia episodes (11.1%), pulmonary infections (6.7%), and wounds (4.4%). This disparity could be attributed to the larger number of urine samples collected in comparison to other clinical specimens. Other studies have also been able to isolate most of their microorganism from urine samples [13,34]. 

Among the seventeen antibiotics used to test antibiotic susceptibility of *K. pneumoniae*, ampicillin had the highest percentage of resistance (100%). These findings agree with the work of Lagha et al. [35], who found *K. pneumoniae* isolates 100% resistant to ampicillin. Followed by ampicillin, we found trimethoprim–sulfamethoxazole, and cefotaxime with 68.9% and 66.7% resistance, respectively. The work of Shadkan et al. [36], in which most of the isolates were resistant to trimethoprim-sulfamethoxazole (52%), and cefotaxime (51%), corroborates our findings. Among the highest rates of resistance, we also reported 66.7% resistance to aztreonam as well as to amoxicillin-clavulanic acid. Not many studies reported increased resistance to aztreonam, and a systematic review and meta-analysis research conducted by Heidary et al. [37] showed that, in Iran, there is a high prevalence of drug-resistant *K. pneumoniae* isolates, with the highest rate of resistance against ampicillin (82.2%), aztreonam (55.4%), and nitrofurantoin (54.5%). Some studies conducted on *K. pneumoniae* have also detected a high prevalence of resistance to amoxicillin-clavulanic acid, such as Kuinkel et al. [13] in 2021, which reported 59.6% of resistance, and Pishtiwan et al. [38], which detected a similar percentage to our own (65%). The latter also reported 100% of susceptibility to amikacin, which is in line with our results for amikacin, which has 84.4% of susceptibility. Among the antibiotics used, tetracycline was also found to be effective (75.6%); however, in a study conducted in Iran, strains isolated from children showed the highest resistance to tetracycline (71.5%), whereas the lowest rate was associated with cefepime (12.7%), imipenem (6%), and gentamicin (6%) [39]. 

Concerning the detection of resistance genes, ESBL-producing *K. pneumoniae* carrying *bla_CTX-M_*, *bla_TEM_*, and *bla_SHV_* genes have been found in clinical samples [40,41]. Similar percentages of combinations between these genes were also found in a study conducted in Egypt [42]. The HS39 isolate only showed resistance to ampicillin, but none of the genes tested was amplified. There are other mechanisms of resistance to ampicillin in *K. pneumoniae*, one possibility is the overexpression of the chromosomal *ampC* gene [43], and another is the presence of non-β-lactam mechanisms of resistance, such as efflux pumps [44] or modifications of penicillin-binding proteins [45].

Regarding aminoglycoside resistance genes, we only detected *aac(3)-II* and *aadA1* genes in our aminoglycoside-resistant strains. Similar results were reported by Mbelle et al. [40], where, besides these two genes, they also detected *aac (6′)−Ib*, *aacA4*, *aadA2*, *aadA5*, *aadA16*, *aph(3′)−Ia*, *strA*, and *strB* genes. Concerning the sample HS125, which was negative for the genes tested, we could find some possibilities why the isolate was negative. For example, the resistance mechanism may involve other genes or mechanisms that we did not test for. There are many other genes that can confer resistance to aminoglycosides, such as *aac(6’)-Ib* [40], *ant(2’’)-Ia* [46], and *aph(3’)-IIIa* [47], which encode for aminoglycoside-modifying enzymes that can modify the structure of aminoglycoside antibiotics, such as amikacin and gentamicin, leading to resistance. Another possibility is the strain may have mutations in the target site of the aminoglycosides, such as 16S rRNA or ribosomal proteins, which can also confer resistance, or also the isolate may have acquired the resistance through a non-genetic mechanism, such as through the production of biofilm or the presence of efflux pumps, which can prevent the drug from reaching its target site [48]. Thus, further research will need to be conducted in order to find out the mechanism underlying aminoglycoside resistance.

Relative to tetracycline resistance genes, we only detected the *tetA* gene in four isolates, and *tetB* was not detected. This result was unexpected, since both genes are commonly detected in clinical samples of *K. pneumoniae* [49,50,51]. As mentioned earlier, in some tetracycline-resistant isolates, we did not detect either *tetA* or *tetB* genes. There are some possible reasons for that, including the acquisition of other tetracycline resistance genes, such as *tetC*, *tetD*, or *tetG* [52], or the developed tetracycline resistance due to mutations in the bacterial ribosome or other cellular components without specific resistance genes [53].

It was unexpected that most of the chloramphenicol-resistant strains were negative for the *cmlA*, *floR*, and *catA* genes, since these are commonly found in clinical strains of *K. pneumoniae* [54]. However, the presence of other chloramphenicol-resistance genes, such as *catB* [55], responsible for enzymatic inactivation, and *cml*, responsible for encoding a chloramphenicol efflux pump that expels chloramphenicol from the cell, or the presence of mutations in the target site of the drug [56], are two of the possible reasons to justify chloramphenicol resistance present in our samples.

To demonstrate the trimethoprim–sulfamethoxazole resistance, we screened the isolates for *sul1*, *sul2*, *sul3*, and *dfrA* genes. The *sul2* gene was the most frequently detected. The same was also reported by Mbelle et al. [40], where they identified *sul2* in 86% of the isolates and *sul1* in 78% of the isolates. In addition to *dfrA* being commonly detected in *K. pneumoniae* strains [40], in our study, it was not detected in any isolate. This could be due to the fact that the strains may have acquired resistance to trimethoprim-sulfamethoxazole through other mechanisms, such as mutations in other genes involved in the folate biosynthesis pathway or through efflux pumps [57]. Given the three isolates that were negative for the trimethoprim-sulfamethoxazole resistance genes evaluated in this work, it is possible that other genes could be conferring resistance to trimethoprim-sulfamethoxazole such as *dfrD*, *dfrG*, *dfrK*, or *sul4* [58,59,60]. Other mechanisms of resistance to trimethoprim-sulfamethoxazole that do not involve the genes listed above may also confer antibiotic resistance, such as alterations in drug uptake or metabolism.

Not detecting *gyrA*, *gyrB*, and *parC* in some quinolone-resistant strains was unforeseen. One possible reason for this is that the primers used for PCR may not have been annealing properly to the target genes due to genetic variation or mutations in the gene sequence, or some strains may have deletions or other genetic alterations in these genes, leading to their absence in the genome. 

In this study, we detected the presence of the *bla_KPC_* gene in all KPC-producing isolates, confirming the phenotypic detection test [61,62,63]. The *bla_PER_* gene is not commonly detected in *K. pneumoniae* strains. However, there have been reports of *bla_PER_*-producing *K. pneumoniae* strains in some countries, such as Iran and Brazil [64,65]. Nevertheless, as far as we know, this is the first report of the *bla_PER_* in *Klebsiella pneumoniae*. Sequencing this isolate is the next step to fully confirm the gene’s presence.

Regarding biofilm production by *K. pneumoniae* strains, we verified that 80% of them were confirmed as biofilm producers. Other studies also reported high rates of biofilm production, such as Türkel et al. [66], who reported that 99% of *K. pneumoniae* isolates were biofilm producers; Seifi et al. [34], who reported 93.6%; and Shadkam et al. [36], who reported 75%. Among the ESBL-producing and KPC-producing *K. pneumoniae*, we verified higher rates of weak biofilm producers, 40.0% and 60.0%, respectively. These results were surprising since the majority of the literature reported more strong biofilm producers among β-lactamase producers [67,68]. However, other studies also reported similar findings to ours [13]. Poovendran et al. [69] found that ESBL-producing strains highly form a biofilm compared with non-ESBL producers. This study did not find any correlation, as the majority of non-ESBL-producing strains were capable of forming a biofilm. This is in accordance with Hasan et al. [70], who reported no correlation between ESBL- and non-ESBL-producing bacteria and their capacity to form biofilms.

## 5. Conclusions

The emergence of *Klebsiella pneumoniae* strains that produce ESBL and KPC has become a significant public health concern. The ability of ESBL- and KPC-producing *K. pneumoniae* to form biofilms is worrisome, as it can facilitate the transmission of antibiotic resistance and prolong infections in healthcare settings. In this study, most of the bacterial strains were found to be multidrug-resistant, with 100% resistance to ampicillin. The most effective antibiotics were also tetracycline and amikacin.

In all ESBL-producing strains, we detected at least one β-lactamase resistance gene, and all KPC-producing isolates had the *bla_KPC_* gene, which confirms the phenotypic detection test. We also detected the *bla_PER_* gene in one KPC-producing isolate. To the best of the author’s knowledge, this is the first time that this gene has been reported in Portugal.

Most of the *K. pneumoniae* strains isolated from hospitalized patients have the capacity for biofilm production. Still, we did not verify an ability to form stronger biofilms by ESBL-producing and KPC-producing *Klebsiella pneumoniae* strains.

Further research is needed to be conducted to better understand the mechanisms involved in the biofilm formation of these isolates by sequencing the strain’s genome and detecting the resistance genes and the virulence genes involved in biofilm production. 

## Figures and Tables

**Figure 1 antibiotics-12-01143-f001:**
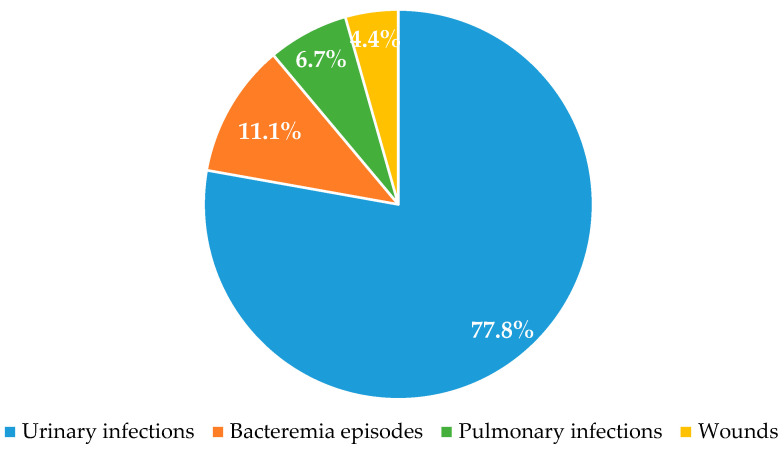
Distribution of *K. pneumoniae* clinical isolates.

**Table 1 antibiotics-12-01143-t001:** Characteristics of *K. pneumoniae* isolates.

Isolate	β-Lactamase	Resistance Profile	Genes Detected
HS10	KPC	AMP AUG FOX CAZ CTX FEP ATM MEM ERT IMP CN CIP NA SXT CHL	*bla_KPC_*, *bla_OXA_*, *bla_CTX-M_*, *bla_SHV_*, *aac(3)-II*, *aadA1*, *sul2*
HS13	KPC	AMP AUG FOX CAZ CTX FEP ATM MEM ERT IMP CN CIP NA SXT CHL	*bla_KPC_*, *bla_OXA_*, *bla_CTX-M_*, *bla_SHV_*, *aac(3)-II*, *aadA1*, *sul2*
HS18	KPC	AMP AUG FOX CAZ CTX FEP ATM ERT IMP CN CIP NA SXT CHL	*bla_KPC_*, *bla_OXA_*, *bla_CTX-M_*, *bla_SHV_*, *aac(3)-II*, *aadA1*, *sul2*
HS24	KPC	AMP AUG FOX CAZ CTX FEP ATM ERT IMP AK CN CIP NA SXT CHL	*bla_KPC_*, *bla_OXA_*, *bla_CTX-M_*, *bla_SHV_*, *aac(3)-II*, *aadA1*, *sul2*
HS74	KPC	AMP AUG FOX CAZ CTX FEP ATM ERT CN CIP NA SXT CHL	*bla_KPC_*, *bla_OXA_*, *bla_CTX-M_*, *bla_SHV_*, *aac(3)-II*, *aadA1*, *gyrB*, *sul2*
HS89	KPC	AMP AUG CTX ATM MEM ERT IMP	*bla_KPC_*, *bla_CTX_*, *bla_TEM_*, *bla_SHV_*
HS99	KPC	AMP AUG CAZ CTX FEP ATM ERT IMP CN CIP NA SXT CHL	*bla_KPC_*, *bla_CTX-M_*, *bla_SHV_*, *aac(3)-II*, *aadA1*, *sul2*
HS102	KPC	AMP AUG FOX CAZ CTX FEP ATM ERT IMP TET CIP NA SXT	*bla_KPC_*, *bla_OXA_*, *bla_CTX_*, *bla_TEM_*, *bla_SHV_*
HS105	KPC	AMP AUG FOX CAZ CTX FEP ATM MEM ERT IMP CN CIP NA SXT CHL	*bla_KPC_*, *bla_OXA_*, *bla_CTX-M_*, *bla_SHV_*, *aadA1*, *sul2*
HS113	KPC	AMP AUG FOX CAZ CTX FEP ATM MEM ERT IMP CN CIP NA SXT CHL	*bla_KPC_*, *bla_OXA_*, *bla_CTX-M_*, *bla_SHV_*, *aac(3)-II*, *aadA1*, *gyrB*, *sul2*
HS125	KPC	AMP AUG FOX CAZ CTX FEP ATM MEM ERT IMP AK CIP NA SXT	*bla_KPC_*, *bla_CTX-M_*, *bla_TEM_*, *bla_SHV_*, *gyrB*, *sul1*
HS128	KPC	AMP AUG FOX CAZ CTX FEP ATM MEM ERT IMP CIP NA SXT	*bla_KPC_*, *bla_CTX-M_*, *bla_TEM_*, *bla_SHV_*, *gyrB*, *sul1*
HS151	KPC	AMP AUG FOX CAZ CTX FEP ATM MEM ERT IMP CN CIP NA SXT CHL	*bla_KPC_*, *bla_OXA_*, *bla_CTX-M_*, *bla_SHV_*, *aac(3)-II*, *aadA1*, *gyrB*, *sul2*
HS153	KPC	AMP AUG FOX CAZ CTX FEP ATM MEM ERT IMP CN CIP NA SXT	*bla_KPC_*, *bla_OXA_*, *bla_CTX-M_*, *bla_SHV_*, *aac(3)-II*, *aadA1*, *gyrB*, *sul2*
HS160	KPC	AMP AUG FOX CAZ CTX FEP ATM ERT IMP CN CIP NA SXT CHL	*bla_KPC_*, *bla_OXA_*, *bla_PER_*, *bla_CTX-M_*, *bla_SHV_*, *aac(3)-II*, *aadA1*, *gyrA*, *gyrB*, *sul2*
HS16	ESBL	AMP AUG CAZ CTX FEP ATM CN CIP NA SXT CHL	*bla_CTX-M_*, *bla_SHV_*, *aac(3)-II*, *aadA1*, *gyrA*, *gyrB*, *sul2*
HS38	ESBL	AMP AUG CAZ CTX FEP ATM CN SXT	*bla_CTX-M_*, *bla_TEM_*, *bla_SHV_*, *aac(3)-II*, *aadA1*, *sul2*
HS72	ESBL	AMP FOX CAZ CTX FEP ATM ERT TET CIP SXT	*bla_CTX-M_*, *bla_TEM_*, *bla_SHV_*, *gyrA*, *gyrB*
HS85	ESBL	AMP AUG CAZ CTX FEP ATM CIP NA SXT CHL	*bla_CTX-M_*, *bla_TEM_*, *bla_SHV_*, *gyrA*, *gyrB*, *parC*
HS97	ESBL	AMP AUG CAZ CTX FEP ATM AK CIP NA SXT CHL	*bla_CTX-M_*, *bla_TEM_*, *bla_SHV_*, *aac(3)-II*, *aadA1*, *cmlA*, *catA*, *gyrA*, *gyrB*, *sul2*, *sul3*
HS98	ESBL	AMP AUG CAZ CTX FEP ATM CN TET CIP NA SXT	*bla_CTX-M_*, *bla_TEM_*, *bla_SHV_*, *aac(3)-II*, *aadA1*, *gyrA*, *gyrB*, *parC*, *sul2*
HS119	ESBL	AMP AUG CAZ CTX FEP ATM CN CIP NA SXT CHL	*bla_CTX-M_*, *bla_SHV_*, *aac(3)-II*, *aadA1*, *gyrB*, *parC*, *sul2*
HS131	ESBL	AMP CTX FEP ATM	*bla_CTX-M_*, *bla_TEM_*, *bla_SHV_*
HS141	ESBL	AMP AUG CAZ CTX FEP ATM AK CN TET CIP SXT	*bla_CTX-M_*, *bla_TEM_*, *bla_SHV_*, *aac(3)-II*, *aadA1*, *tetA*, *gyrA*, *gyrB*, *parC*, *sul2*
HS143	ESBL	AMP AUG FOX CAZ CTX FEP ATM CN CIP NA SXT CHL	*bla_CTX-M_*, *bla_SHV_*, *aac(3)-II*, *aadA1*, *gyrA*, *gyrB*, *parC*, *sul2*
HS147	ESBL	AMP AUG CAZ CTX FEP ATM CN CIP NA SXT CHL	*bla_CTX-M_*, *bla_SHV_*, *aac(3)-II*, *aadA1*, *gyrA*, *gyrB*, *sul2*
HS149	ESBL	AMP AUG CAZ CTX FEP ATM CN TET CIP NA SXT	*bla_CTX-M_*, *bla_TEM_*, *bla_SHV_*, *aac(3)-II*, *aadA1*, *gyrB*, *parC*, *sul2*
HS154	ESBL	AMP AUG CAZ CTX FEP ATM CIP NA SXT CHL	*bla_CTX-M_*, *bla_SHV_*, *gyrB*, *parC*, *sul1*, *sul2*
HS161	ESBL	AMP AUG CAZ CTX FEP ATM CN CIP NA SXT CHL	*bla_CTX-M_*, *bla_SHV_*, *aadA1*, *gyrB*, *parC*, *sul2*
HS163	ESBL	AMP AUG CAZ CTX FEP ATM CN CIP NA SXT CHL	*bla_CTX-M_*, *aac(3)-II*, *aadA1*, *gyrB*, *sul2*
HS26	-	AMP AUG	*bla_SHV_*
HS32	-	AMP CN	*bla_TEM_*, *bla_SHV_*, *aadA1*
HS33	-	AMP CN	*bla_TEM_*, *bla_SHV_*, *aadA1*
HS35	-	AMP CN	*bla_TEM_*, *bla_SHV_*, *aac(3)-II*, *aadA1*
HS39	-	AMP	-
HS40	-	AMP TET CIP SXT CHL	*bla_SHV_*, *tetA*, *gyrB*, *sul2*
HS41	-	AMP CN	*bla_TEM_*, *bla_SHV_*, *aac(3)-II*, *aadA1*
HS43	-	AMP TET SXT	*bla_TEM_*, *bla_SHV_*, *tetA*, *sul1*
HS46	-	AMP AK	*bla_TEM_*, *bla_SHV_*, *aac(3)-II*, *aadA1*
HS47	-	AMP AK CN	*bla_TEM_*, *bla_SHV_*, *aac(3)-II*, *aadA1*
HS49	-	AMP CN	*bla_TEM_*, *bla_SHV_*, *aadA1*
HS53	-	AMP AUG AK TET CIP SXT	*bla_TEM_*, *bla_SHV_*, *aac(3)-II*, *aadA1*, *tetA*, *gyrB*, *sul1*
HS60	-	AMP CN	*bla_TEM_*, *bla_SHV_*, *aac(3)-II*, *aadA1*
HS62	-	AMP CN	*bla_TEM_*, *bla_SHV_*, *aac(3)-I*,*I aadA1*
HS63	-	AMP	*bla_TEM_*, *bla_SHV_*

**Table 2 antibiotics-12-01143-t002:** Antibiotic susceptibility rates of *Klebsiella pneumoniae* (*n* = 45).

Antibiotics	*Klebsiella pneumoniae* (*n* = 45)		
	Resistant *n* (%)	Intermediate *n* (%)	Sensitive *n* (%)
Ampicillin	45 (100.0)	0 (0.0)	0 (0.0)
Trimethoprim-sulfamethoxazole	31 (68.9)	0 (0.0)	14 (31.1)
Amoxicillin-clavulanic acid	30 (66.7)	0 (0.0)	15 (33.3)
Cefotaxime	30 (66.7)	0 (0.0)	15 (33.3)
Aztreonam	30 (66.7)	0 (0.0)	15 (33.3)
Cefepime	29 (64.4)	0 (0.0)	16 (35.6)
Ciprofloxacin	29 (64.4)	0 (0.0)	16 (35.6)
Gentamicin	29 (64.4)	0 (0)	16 (35.6)
Ceftazidime	28 (62.2)	2 (4.4)	15 (33.3)
Nalidixic acid	25 (55.6)	1 (2.2)	19 (42.2)
Chloramphenicol	20 (44.4)	0 (0.0)	25 (55.6)
Ertapenem	16 (35.6)	0 (0.0)	29 (64.4)
Cefoxitin	15 (33.3)	0 (0.0)	30 (66.7)
Imipenem	14 (31.1)	1 (2.2)	30 (66.7)
Meropenem	9 (20.0)	5 (11.1)	31 (68.9)
Tetracycline	8 (17.8)	3 (6.7)	34 (75.6)
Amikacin	7 (15.6)	0(0.0)	38 (84.4)

**Table 3 antibiotics-12-01143-t003:** Biofilm production among *Klebsiella pneumoniae* isolates in relation to clinical specimen and type of β-lactamase producer.

	Biofilm Producers			Non-Biofilm Producers
	Weak *n* (%)	Moderate *n* (%)	Strong *n* (%)	Non-Producers *n* (%)
*Klebsiella pneumoniae* (*n* = 45)	26 (57.8)	9 (20.0)	1 (2.2)	9 (20.0)
**Clinical Specimen**				
Urinary infections (*n* = 35)	21 (60.0)	6 (17.1)	1 (2.9)	7 (20.0)
Bacteremia episodes (*n* = 5)	2 (40.0)	2 (40.0)	0 (0.0)	1 (20.0)
Pulmonary infections (*n* = 3)	3 (100.0)	0 (0.0)	0 (0.0)	0 (0.0)
Wounds (*n* = 2)	0 (0.0)	1 (50.0)	0 (0.0)	1 (50.0)
**Type of β-lactamase producer**				
ESBL producer (*n* = 15)	6 (40.0)	4 (26.7)	1 (6.7)	4 (26.7)
KPC producer (*n* = 15)	9 (60.0)	2 (13.3)	0 (0.0)	4 (26.7)
Non- β-lactamese producer (*n* = 15)	11 (73.3)	3 (20.0)	0 (0.0)	1 (6.7)

## Data Availability

Not applicable.

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
