# Peer review of "Evaluation of the Ability to Form Biofilms in KPC-Producing and ESBL-Producing Klebsiella pneumoniae Isolated from Clinical Samples"

_antibiotics, 2023, doi:10.3390/antibiotics12071143_

Round 1
Reviewer 1 Report
The submitted manuscript is almost fully acceptable. I have only three minor comments.
Lines 175-177: "The phenotypic profile of the 45 Klebsiella pneumoniae isolates demonstrated that 73% of the strains were multi-resistant, showing resistance to three or more antibiotic classes (Table 1)." Actually, in Table 1 are displayed the numbers of resistant isolates to every single antibiotic taken into consideration, while the multi-resistance profile is reported in Table 2. Please, correct.
Line 287: "Among the seventeen antibiotics used to test antibiotic susceptibility against....", either replace the word "susceptibility" with "efficacy", or replace the word "against" with "of".
Lines 392-393: "We also detected in one KPC-producing isolate the blaPER gene, and we belive is the first time being reported in Portugal." Please, delete this statement because it is a repetition of the previous one on lines 391-392.
Reviewer 2 Report
While the results are not too exciting, it is an important negative result. However, it is a worthwhile catalog of the different strains.
Only small improvements need to be checked with the use of English in the paper. These did not interfere with the readability of the article.
Reviewer 3 Report
This study investigated beta-lactamase-producing and biofilm-producing of Klebsiella pneumoniae strains isolated in a hospital. The topic of this study is important for control of nosocomial infection.
This study is well designed, and the results obtained from this study are reliable. However, the authors should revise their manuscript according to the below comments.
[Major comment]
The authors investigated the bacteriological characteristics (beta-lactamase-producing and biofilm-producing) of the Klebsiella pneumoniae strains in detail. However, this manuscript does not contain enough data about the patients from which the strains were isolated. The authors should analyze the relationship between the bacteriological characteristics of the strains and the patients, and the results of the analysis should be added to their manuscript.
[Minor comment]
Abbreviations should not be used in a title.
Round 2
Reviewer 3 Report
I understood the limitations of this study.
In addition, I also understood that the authors cannot resolve the limitations due to the ethical concern.
Though the limitations, the results of this study are valuable in publication in Antibiotics.